# Segmenting Natural Language Sentences via Lexical Unit Analysis

## Abstract

In this work, we present **L**exical **U**nit **A**nalysis (**LUA**), a framework for general sequence segmentation tasks. Given a natural language sentence, LUA scores all the valid segmentation candidates and utilizes dynamic programming (DP) to extract the maximum scoring one. LUA enjoys a number of appealing properties such as inherently guaranteeing the predicted segmentation to be valid and facilitating globally optimal training and inference. Besides, the practical time complexity of LUA can be reduced to linear time, which is very efficient. We have conducted extensive experiments on 5 tasks, including syntactic chunking, named entity recognition (NER), slot filling, Chinese word segmentation, and Chinese part-of-speech (POS) tagging, across 15 datasets. Our models have achieved the state-of-the-art performances on 13 of them. The results also show that the F1 score of identifying long-length segments is notably improved.

## 1 Introduction

Sequence segmentation is essentially the process of partitioning a sequence of fine-grained lexical units into a sequence of coarse-grained ones. In some scenarios, each composed unit is assigned a categorical label. For example, Chinese word segmentation splits a character sequence into a word sequence (Xue, 2003). Syntactic chunking segments a word sequence into a sequence of labeled groups of words (i.e., constituents) (Sang & Buchholz, 2000).

There are currently two mainstream approaches to sequence segmentation. The most common is to regard it as a sequence labeling problem by using IOB tagging scheme (Mesnil et al., 2014; Ma & Hovy, 2016; Liu et al., 2019b; Chen et al., 2019a; Luo et al., 2020). A representative work is Bidirectional LSTM-CRF (Huang et al., 2015), which adopts LSTM (Hochreiter & Schmidhuber, 1997) to read an input sentence and CRF (Lafferty et al., 2001) to decode the label sequence. This type of method is very effective, providing tons of state-of-the-art performances. However, it is vulnerable to producing invalid labels, for instance, "O, I-tag, I-tag". This problem is very severe in low resource settings (Peng et al., 2017). In experiments (see section 4.6), we also find that it performs poorly in recognizing long-length segments.

Recently, there is a growing interest in span-based models (Zhai et al., 2017; Li et al., 2019; Yu et al., 2020). They treat a span rather than a token as the basic unit for labeling. Li et al. (2019) cast named entity recognition (NER) to a machine reading comprehension (MRC) task, where entities are extracted as retrieving answer spans. Yu et al. (2020) rank all the spans in terms of the scores predicted by a bi-affine model (Dozat & Manning, 2016). In NER, span-based models have significantly outperformed their sequence labeling based counterparts. While these methods circumvent the use of IOB tagging scheme, they still rely on post-processing rules to guarantee the extracted span set to be valid. Moreover, since span-based models are locally normalized at span level, they potentially suffer from the label bias problem (Lafferty et al., 2001).

This paper seeks to provide a new framework which infers the segmentation of a unit sequence by directly selecting from all valid segmentation candidates, instead of manipulating tokens or spans. To this end, we propose **L**exical **U**nit **A**nalysis (**LUA**) in this paper. LUA assigns a score to every valid segmentation candidate and leverages dynamic programming (DP) (Bellman, 1966) to search for the maximum scoring one. The score of a segmentation is computed by using the scores of its all segments. Besides, we adopt neural networks to score every segment of the input sentence.

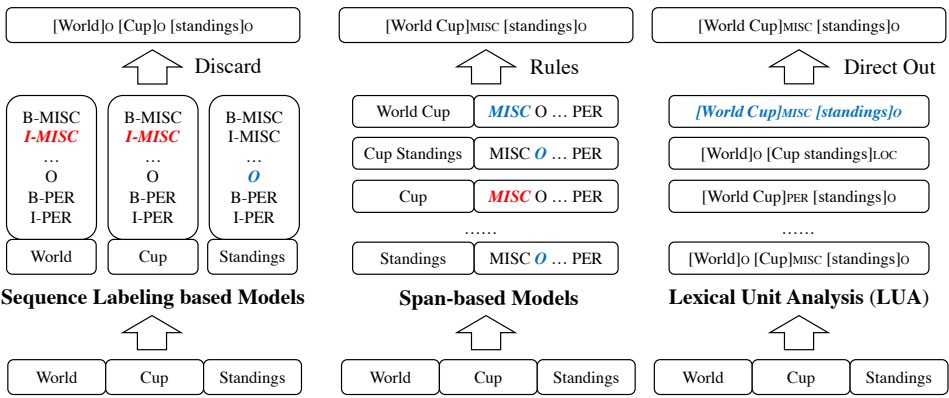

Figure 1: A toy example to show LUA and how it differs from prior methods. The items in blue and red respectively denote valid and invalid predictions.

The purpose of using DP is to solve the intractability of extracting the maximum scoring segmentation candidate by brute-force search. The time complexity of LUA is quadratic time, yet it can be optimized to linear time in practice by performing parallel matrix computations. For training criterion, we incur a hinge loss between the ground truth and the predictions. We also extend LUA to unlabeled segmentation and capturing label correlations.

Figure 1 illustrates the comparison between previous methods and the proposed LUA. Prior models at token level and span level are vulnerable to generating invalid predictions, and hence rely on heuristic rules to fix them. For example, in the middle part of Figure 1, the spans of two inferred named entities, $[\text{Word Cup}]_{\text{MISC}}$ and $[\text{Cup}]_{\text{MISC}}$, conflicts, which is mitigated by comparing the predicted scores. LUA scores all possible segmentation candidates and uses DP to extract the maximum scoring one. In this way, our models guarantee the predictions to be valid. Moreover, the globality of DP addresses the label bias problem.

Extensive experiments are conducted on syntactic chunking, NER, slot filling, Chinese word segmentation, and Chinese part-of-speech (POS) tagging across 15 tasks. We have obtained new state-of-the-art results on 13 of them and performed competitively on the others. In particular, we observe that LUA is expert at identifying long-length segments.

## 2 METHODOLOGY

We denote an input sequence (i.e., fine-grained lexical units) as $\mathbf{x} = [x_1, x_2, \cdots, x_n]$, where $n$ is the sequence length. An output sequence (i.e., coarse-grained lexical units) is represented as the segmentation $\mathbf{y} = [y_1, y_2, \cdots, y_m]$ with each segment $y_k$ being a triple $(i_k, j_k, t_k)$. $m$ denotes its length. $(i_k, j_k)$ specifies a span that corresponds to the phrase $\mathbf{x}_{i_k, j_k} = [x_{i_k}, x_{i_k+1}, \cdots, x_{j_k}]$. $t_k$ is a label from the label space $\mathcal{L}$. We define a valid segmentation candidate as its segments are non-overlapping and fully cover the input sequence.

A case extracted from CoNLL-2003 dataset (Sang & De Meulder, 2003):

$$\mathbf{x} = [[\text{SOS}], \text{Sangthai}, \text{Glory}, 22/11/96, 3000, \text{Singapore}]$$
$$\mathbf{y} = [(1, 1, \text{O}), (2, 3, \text{MISC}), (4, 4, \text{O}), (5, 5, \text{O}), (6, 6, \text{LOC})]$$

Start-of-sentence symbol [SOS] is added in the pre-processing stage.

### 2.1 MODEL: SCORING SEGMENTATION CANDIDATES

We denote $\mathcal{Y}$ as the universal set that contains all valid segmentation candidates. Given one of its members $\mathbf{y} \in \mathcal{Y}$, we compute the score $f(\mathbf{y})$ as

$$f(\mathbf{y}) = \sum_{(i,j,t) \in \mathbf{y}} \left( s_{i,j}^c + s_{i,j,t}^l \right), \tag{1}$$

---

**Algorithm 1:** Inference via Dynamic Programming (DP)

---

**Input:** Composition score $s_{i,j}^c$ and label score $s_{i,j,t}^l$ for every possible segment $(i, j, t)$.
**Output:** The maximum segmentation scoring candidate $\hat{\mathbf{y}}$ and its score $f(\hat{\mathbf{y}})$.

1 Set two $n \times n$ shaped matrices, $\mathbf{c}^L$ and $\mathbf{b}^c$, for computing maximum scoring labels.
2 Set two $n$-length vectors, $\mathbf{g}$ and $\mathbf{b}^g$, for computing maximum scoring segmentation.
3 **for** $1 \leq i \leq j \leq n$ **do**
4      Compute the maximum label score for each span $(i, j)$: $s_{i,j}^L = \max_{t \in \mathcal{L}} s_{i,j,t}^l$.
5      Record the backtracking index: $b_{i,j}^c = \arg\max_{t \in \mathcal{L}} s_{i,j,t}^l$.
6 Initialize the value of the base case $\mathbf{x}_{1,1}$: $g_1 = s_{1,1}^c + s_{1,1}^L$.
7 **for** $i \in [2, 3, \cdots, n]$ **do**
8      Compute the value of the prefix $\mathbf{x}_{1,i}$: $g_i = \max_{1 \leq j \leq i-1} \left( g_{i-j} + (s_{i-j+1,i}^c + s_{i-j+1,i}^L) \right)$.
9      Record the backtracking index: $b_i^g = \arg\max_{1 \leq j \leq i-1} \left( g_{i-j} + (s_{i-j+1,i}^c + s_{i-j+1,i}^L) \right)$.
10 Get the maximum scoring candidate $\hat{\mathbf{y}}$ by back tracing the tables $\mathbf{b}^g$ and $\mathbf{b}^c$.
11 Get the maximum segmentation score: $f(\hat{\mathbf{y}}) = g_n$.

---

where $s_{i,j}^c$ is the composition score to estimate the feasibility of merging several fine-grained units $[x_i, x_{i+1}, \cdots, x_j]$ into a coarse-grained unit and $s_{i,j,t}^l$ is the label score to measure how likely the label of this segment is $t$. Both scores are obtained by a scoring model.

**Scoring Model.** a scoring model scores all possible segments $(i, j, t)$ for an input sentence $\mathbf{x}$. Firstly, we get the representation for each fine-grained unit. Following prior works (Li et al., 2019; Luo et al., 2020; Yu et al., 2020), we adopt BERT (Devlin et al., 2018), a powerful pre-trained language model, as the sentence encoder. Specifically, we have

$$[\mathbf{h}_1^w, \mathbf{h}_2^w \cdots, \mathbf{h}_n^w] = \text{BERT}(\mathbf{x}), \tag{2}$$

Then, we compute the representation for a coarse-grained unit $\mathbf{x}_{i,j}, 1 \leq i \leq j \leq n$ as

$$\mathbf{h}_{i,j}^p = \mathbf{h}_i^w \oplus \mathbf{h}_j^w \oplus (\mathbf{h}_i^w - \mathbf{h}_j^w) \oplus (\mathbf{h}_i^w \odot \mathbf{h}_j^w), \tag{3}$$

where $\oplus$ is vector concatenation and $\odot$ is element-wise product.

Eventually, we employ two non-linear feedforward networks to score a segment $(i, j, t)$:

$$s_{i,j}^c = \left(\mathbf{v}^c\right)^T \tanh(\mathbf{W}^c \mathbf{h}_{i,j}^p), \quad s_{i,j,t}^l = \left(\mathbf{v}_t^l\right)^T \tanh(\mathbf{W}^l \mathbf{h}_{i,j}^p), \tag{4}$$

where $\mathbf{v}^c$, $\mathbf{W}^c$, $\mathbf{v}_t^l, t \in \mathcal{L}$, and $\mathbf{W}^l$ are all learnable parameters. Besides, the scoring model used here can be flexibly replaced by any regression method.

## 2.2 INFERENCE VIA DYNAMIC PROGRAMMING

The prediction of the maximum scoring segmentation candidate can be formulated as

$$\hat{\mathbf{y}} = \arg\max_{\mathbf{y} \in \mathcal{Y}} f(\mathbf{y}). \tag{5}$$

Because the size of search space $|\mathcal{Y}|$ increases exponentially with respect to the sequence length $n$, brute-force search to solve Equation 5 is computationally infeasible. LUA uses DP to address this issue, which is facilitated by the decomposable nature of Equation 1.

DP is a well-known optimization method which solves a complicated problem by breaking it down into simpler sub-problems in a recursive manner. The relation between the value of the larger problem and the values of its sub-problems is called the Bellman equation.

**Sub-problem.** In the context of LUA, the sub-problem of segmenting an input unit sequence $\mathbf{x}$ is segmenting its prefixes $\mathbf{x}_{1,i}, 1 \leq i \leq n$. We define $g_i$ as the maximum segmentation score of the prefix $\mathbf{x}_{1,i}$. Under this scheme, we have $\max_{\mathbf{y} \in \mathcal{Y}} f(\mathbf{y}) = g_n$.

**The Bellman Equation.** The relatinship between segmenting a sequence $\mathbf{x}_{1,i}, i > 1$ and segmenting its prefixes $x_{1,i-j}, 1 \leq j \leq i-1$ is built by the last segments $(i-j+1, i, t)$:

$$g_i = \max_{1 \leq j \leq i-1} \left( g_{i-j} + (s^c_{i-j+1,i} + \max_{t \in \mathcal{L}} s^l_{i-j+1,i,t}) \right). \tag{6}$$

In practice, to reduce the time complexity of above equation, the last term is computed beforehand as $s^L_{i,j} = \max_{t \in \mathcal{L}} s^l_{i,j,t}, 1 \leq i \leq j \leq n$. Hence, Equation 6 is reformulated as

$$g_i = \max_{1 \leq j \leq i-1} \left( g_{i-j} + (s^c_{i-j+1,i} + s^L_{i-j+1,i}) \right). \tag{7}$$

The base case is the first token $\mathbf{x}_{1,1} = [[\text{SOS}]]$. We get its score $g_1$ as $s^c_{1,1} + s^L_{1,1}$.

Algorithm 1 shows how DP is applied in inference. Firstly, we set two matrices and two vectors to store the solutions to the sub-problems (1-st to 2-nd lines). Secondly, we get the maximum label scores for all the spans (3-rd to 5-th lines). Then, we initialize the trivial case $g_1$ and recursively calculate the values for prefixes $\mathbf{x}_{1,i}, i > 1$ (6-th to 9-th lines). Finally, we get the predicted segmentation $\hat{\mathbf{y}}$ and its score $f(\hat{\mathbf{y}})$ (10-th to 11-th lines).

The time complexity of Algorithm 1 is $\mathcal{O}(n^2)$. By performing the $\max$ operation of Equation 7 in parallel on GPU, it can be optimized to only $\mathcal{O}(n)$, which is highly efficient. Besides, DP, as the backbone of the proposed model, is non-parametric. The trainable parameters only exist in the scoring model part. These show LUA is a very light-weight algorithm.

## 2.3 Training Criterion

We adopt max-margin penalty as the loss function for training. Given the predicted segmentation $\hat{\mathbf{y}}$ and the ground truth segmentation $\mathbf{y}^*$, we have

$$\mathcal{J} = \max \left( 0, 1 - f(\mathbf{y}^*) + f(\hat{\mathbf{y}}) \right). \tag{8}$$

## 3 Extensions of LUA

We propose two extensions of LUA for generalizing it to different scenarios.

**Unlabeled Segmentation.** In some tasks (e.g., Chinese word segmentation), the segments are unlabeled. Under this scheme, the Equation 1 and Equation 7 are reformulated as

$$f(\mathbf{y}) = \sum_{(i,j) \in \mathbf{y}} s^c_{i,j}, \quad g_i = \max_{1 \leq j \leq i-1} (g_{i-j} + s^c_{i-j+1,i}). \tag{9}$$

**Capturing Label Correlations.** In some tasks (e.g., syntactic chunking), the labels of segments are strongly correlated. To incorporate this information, we redefine $f(\mathbf{y})$ as

$$f(\mathbf{y}) = \sum_{1 \leq k \leq m} \left( s^c_{i_k,j_k} + s^l_{i_k,j_k,t_k} \right) + \sum_{1 \leq k \leq m} s^d_{t_{k-q+1}, t_{k-q+2}, \cdots, t_k}. \tag{10}$$

Score $s^d_{t_{k-q+1}, t_{k-q+2}, \cdots, t_k}$ models the label dependencies among $q$ successive segments, $\mathbf{y}_{k-q+1,k}$. In practice, we find $q = 2$ balances the efficiency and the effectiveness well, and thus parameterize a learnable matrix $\mathbf{W}^d \in \mathbb{R}^{|\mathcal{V}| \times |\mathcal{V}|}$ to implement it.

The corresponding Bellman equation to above scoring function is

$$g_{i,t} = \max_{1 \leq j \leq i-1} \left( \max_{t' \in \mathcal{L}} (g_{i-j,t'} + s^d_{t',t}) + (s^c_{i-j+1,i} + s^l_{i-j+1,i,t}) \right), \tag{11}$$

where $g_{i,t}$ is the maximum score of labeling the last segment of the prefix $\mathbf{x}_{1,i}$ with $t$. For initialization, we set the value of $g^d_{1,\text{O}}$ as 0 and the others as $-\infty$. By performing the inner loops of two $\max$ operations in parallel, the practical time complexity for computing $g_{i,t}, 1 \leq i \leq n, t \in \mathcal{L}$ is also $\mathcal{O}(n)$. Ultimately, the segmentation score $f(\hat{\mathbf{y}})$ is obtained by $\max_{t \in \mathcal{L}} g_{n,t}$.

This extension further improves the results on syntactic chunking and Chinese POS tagging, as both tasks have rich sequential features among the labels of segments.

| Model | AS | MSR | CITYU | PKU | CTB6 |
|---|---|---|---|---|---|
| Rich Pretraining (Yang et al., 2017) | 95.7 | 97.5 | 96.9 | 96.3 | 96.2 |
| Bi-LSTM (Ma et al., 2018) | 96.2 | 98.1 | 97.2 | 96.1 | 96.7 |
| Multi-Criteria Learning + BERT (Huang et al., 2019) | 96.6 | 97.9 | 97.6 | 96.6 | 97.6 |
| BERT (Meng et al., 2019) | 96.5 | 98.1 | 97.6 | 96.5 | - |
| Glyce + BERT (Meng et al., 2019) | 96.7 | **98.3** | 97.9 | 96.7 | - |
| Unlabeled LUA | **96.94** | 98.27 | **98.21** | **96.88** | **98.13** |

Table 1: Experiment results on Chinese word segmentation.

| Model | | CTB5 | CTB6 | CTB9 | UD1 |
|---|---|---|---|---|---|
| Bi-RNN + CRF (Single) (Shao et al., 2017) | | 94.07 | 90.81 | 91.89 | 89.41 |
| Bi-RNN + CRF (Ensemble) (Shao et al., 2017) | | 94.38 | - | 92.34 | 89.75 |
| Lattice-LSTM (Meng et al., 2019) | | 95.14 | 91.43 | 92.13 | 90.09 |
| Glyce + Lattice-LSTM (Meng et al., 2019) | | 95.61 | 91.92 | 92.38 | 90.87 |
| BERT (Meng et al., 2019) | | 96.06 | 94.77 | 92.29 | 94.79 |
| Glyce + BERT (Meng et al., 2019) | | 96.61 | 95.41 | 93.15 | 96.14 |
| This Work | LUA | 96.79 | 95.39 | 93.22 | 96.01 |
| | LUA w/ Label Correlations | **97.96** | **96.63** | **93.95** | **97.08** |

Table 2: Experiment results on the four datasets of Chinese POS tagging.

# 4 EXPERIMENTS

We have conducted extensive studies on 5 tasks, including Chinese word segmentation, Chinese POS tagging, syntactic chunking, NER, and slot filling, across 15 datasets. Firstly, Our models have achieved new state-of-the-art performances on 13 of them. Secondly, the results demonstrate that the F1 score of identifying long-length segments has been notably improved. Lastly, we show that LUA is a very efficient algorithm concerning the running time.

## 4.1 SETTINGS

We use the same configurations for all 15 datasets. L2 regularization and dropout ratio are respectively set as $1 \times 10^{-6}$ and 0.2 for reducing overfit. We use Adam (Kingma & Ba, 2014) to optimize our model. Following prior works, $BERT_{BASE}$ is adopted as the sentence encoder. We use uncased $BERT_{BASE}$ for slot filling, Chinese $BERT_{BASE}$ for Chinese tasks (e.g., Chinese POS tagging), and cased $BERT_{BASE}$ for others (e.g., syntactic chunking). In addition, the improvements of our model over baselines are statistically significant with $p < 0.05$ under t-test.

## 4.2 CHINESE WORD SEGMENTATION

Chinese word segmentation splits a Chinese character sequence into a sequence of Chinese words. We use SIGHAN 2005 bake-off (Emerson, 2005) and Chinese Treebank 6.0 (CTB6) (Xue et al., 2005). SIGHAN 2005 back-off consists of 5 datasets, namely AS, MSR, CITYU, and PKU. Following Ma et al. (2018), we randomly select 10% training data as development set. We convert all digits, punctuation, and Latin letters to half-width for handling full/half-width mismatch between training and test set. We also convert AS and CITYU to simplified Chinese. For CTB6, we follow the same format and partition as in Yang et al. (2017); Ma et al. (2018).

Table 1 depicts the experiment results. All the results of baselines are from Yang et al. (2017); Ma et al. (2018); Huang et al. (2019); Meng et al. (2019). We have achieved new state-of-the-art performance on all datasets except MSR. Our model improves the F1 score by 0.25% on AS, 0.32% on CITYU, 0.19% on PKU, and 0.54% on CTB6. Note that our model doesn't use any external resources, such as glyph information (Meng et al., 2019) or POS tags (Yang et al., 2017). Despite this, our model is still competitive with Glyce + BERT on MSR.

## 4.3 CHINESE POS TAGGING

Chinese POS tagging jointly segments a Chinese character sequence and assigns a POS tag to each segmented unit. We use Chinese Treebank 5.0 (CTB5), CTB6, Chinese Treebank 9.0 (CTB9) (Xue

| Model | | Chunking | NER | |
|---|---|---|---|---|
| | | CoNLL-2000 | CoNLL-2003 | OntoNotes 5.0 |
| Bi-LSTM + CRF (Huang et al., 2015) | | 94.46 | 90.10 | - |
| Flair Embeddings (Akbik et al., 2018) | | 96.72 | 93.09 | 89.3 |
| GCDT w/ BERT (Liu et al., 2019b) | | 96.81 | 93.23 | - |
| BERT-MRC (Li et al., 2019) | | - | 93.04 | 91.11 |
| HCR w/ BERT (Luo et al., 2020) | | - | 93.37 | 90.30 |
| BERT-Biaffine Model (Yu et al., 2020) | | - | **93.5** | 91.3 |
| This Work | LUA | 96.95 | 93.46 | **92.09** |
| | LUA w/ Label Correlations | **97.23** | - | - |

Table 3: Experiment results on syntactic chunking and NER.

et al., 2005), and the Chinese section of Universal Dependencies 1.4 (UD1) (Nivre et al., 2016). CTB5 is comprised of newswire data. CTB9 consists of source texts in various genres, which cover CTB5. we convert the texts in UD1 from traditional Chinese into simplified Chinese. We follow the same train/dev/test split for above datasets as in Shao et al. (2017).

Table 2 shows the experiment results. The performances of all baselines are reported from Meng et al. (2019). Our model LUA w/ Label Correlations has yielded new state-of-the-art results on all the datasets: it improves the F1 scores by $1.35\%$ on CTB5, $1.22\%$ on CTB6, $0.8\%$ on CTB9, and $0.94\%$ on UD1. Moreover, the basic LUA without capturing the label correlations also outperforms the strongest baseline, Glyce + BERT, by $0.18\%$ on CTB5 and $0.07\%$ on CTB9. All these facts further verify the effectiveness of LUA and its extension.

## 4.4 SYNTACTIC CHUNKING AND NER

Syntactic chunking aims to find phrases related to syntatic category for a sentence. We use CoNLL-2000 dataset (Sang & Buchholz, 2000), which defines 11 syntactic chunk types (NP, VP, PP, etc.) and follow the standard splittings of training and test datasets as previous work. NER locates the named entities mentioned in unstructured text and meanwhile classifies them into predefined categories. We use CoNLL-2003 dataset (Sang & De Meulder, 2003) and OntoNotes 5.0 dataset (Pradhan et al., 2013). CoNLL-2003 dataset consists of 22137 sentences totally and is split into 14987, 3466, and 3684 sentences for the training set, development set, and test set, respectively. It is tagged with four linguistic entity types (PER, LOC, ORG, MISC). OntoNotes 5.0 dataset contains 76714 sentences from a wide variety of sources (e.g., magazine and newswire). It includes 18 types of named entity, which consists of 11 types (Person, Organization, etc.) and 7 values (Date, Percent, etc.). We follow the same format and partition as in Li et al. (2019); Luo et al. (2020); Yu et al. (2020). In order to fairly compare with previous reported results, we convert the predicted segments into IOB format and utilize conlleval script[1] to compute the F1 score at test time.

Table 3 shows the results. Most of baselines are directly taken from Akbik et al. (2018); Li et al. (2019); Luo et al. (2020); Yu et al. (2020). Besides, following Luo et al. (2020), we rerun the source code[2] of GCDT and report its result on CoNLL-2000 with standard evaluation method. Generally, our proposed models LUA w/o Label Correlations yield competitive performance over state-of-the-art models on both Chunking and NER tasks. Specifically, regarding to the NER task, on CoNLL-2003 dataset our model LUA outperforms several strong baselines including Flair Embedding, and it is comparable to the state-of-the-art model (i.e., BERT-Biaffine Model). In particular, on OntoNotes dataset, LUA outperforms it by $0.79\%$ points and establishes a new state-of-the-art result. Regarding to the Chunking task, LUA advances the best model (GCDT) and the improvements are further enlarged to $0.42\%$ points by LUA w/ Label Correlations.

## 4.5 SLOT FILLING

Slot filling, as an important task in spoken language understanding (SLU), extracts semantic constituents from an utterance. We use ATIS dataset (Hemphill et al., 1990), SNIPS dataset (Coucke et al., 2018), and MTOD dataset (Schuster et al., 2018). ATIS dataset consists of audio recordings of

---

[1]https://www.clips.uantwerpen.be/conll2000/chunking/conlleval.txt.
[2]https://github.com/Adaxry/GCDT.

| Model | ATIS | SNIPS | MTOD |
|---|---|---|---|
| Slot-Gated SLU (Goo et al., 2018) | 95.20 | 88.30 | 95.12 |
| Bi-LSTM + EMLo (Siddhant et al., 2019) | 95.42 | 93.90 | - |
| Joint BERT (Chen et al., 2019b) | 96.10 | 97.00 | 96.48 |
| CM-Net (Liu et al., 2019c) | 96.20 | 97.15 | - |
| This Work — LUA | 96.15 | 97.10 | 97.53 |
| This Work — LUA w/ Intent Detection | **96.27** | **97.20** | **97.55** |

Table 4: Experiment results on the three datasets of slot filling.

| Model | $1-3$ (8695) | $4-7$ (2380) | $8-11$ (151) | $12-24$ (31) | Overall |
|---|---|---|---|---|---|
| HCR w/ BERT | 91.15 | 85.22 | 50.43 | 20.67 | 90.27 |
| BERT-Biaffine Model | 91.67 | 87.23 | 70.24 | 40.55 | 91.26 |
| LUA | **92.31** | **88.52** | **77.34** | **57.27** | **92.09** |

Table 5: The F1 scores for NER models on different segment lengths. $A - B(N)$ denotes that there are $N$ entities whose span lengths are between $A$ and $B$.

people making flight reservations. The training set contains 4478 utterances and the test set contains 893 utterances. SNIPS dataset is collected by Snips personal voice assistant. The training set contains 13084 utterances and the test set contains 700 utterances. MTOD dataset has three domains, including Alarm, Reminder, and Weather. We use the English part of MTOD dataset, where training set, dev set, and test set respectively contain 30521, 4181, and 8621 utterances. We follow the same partition of above datasets as in Goo et al. (2018); Schuster et al. (2018).

Table 4 summarizes the experiment results for slot filling. On ATIS and SNIPS, we take the results of all baselines as reported in Liu et al. (2019c) for comparison. On MTOD, we rerun the open source toolkits, Slot-gated SLU[3] and Joint BERT[4]. As all previous approaches jointly model slot filling and intent detection (a classification task in SLU), we follow them to augment LUA with intent detection for a fair comparison. As shown in Table 4, the augmented LUA has surpassed all baselines and obtained state-of-the-art results on the three datasets: it increases the F1 scores by around $0.05\%$ on ATIS and SNIPS, and delivers a substantial gain of $1.11\%$ on MTOD. It's worth mentioning that LUA even outperforms the strong baseline Joint BERT with a margin of $0.18\%$ and $0.21\%$ on ATIS and SNIPS without modeling intent detection.

### 4.6 LONG-LENGTH SEGMENT IDENTIFICATION

Since LUA doesn't resort to IOB tagging scheme, it should be more accurate in recognizing long-length segments than prior methods. To verify this intuition, we evaluate different models on the segments of different lengths. This study is investigated on OntoNotes 5.0 dataset. Two strong models are adopted as the baselines: one is the best sequence labeling model (i.e., HCR) and the other is the best span-based model (i.e., BERT-Biaffine Model). Both baselines are reproduced by rerunning their open source codes, biaffine-ner[5] and Hire-NER[6].

The results are shown in Table 5. On the one hand, both LUA and Biaffine Model obtain much higher scores of extracting long-length entities than HCR. For example, LUA outperforms HCR w/ BERT by almost twofold on range $12 - 24$. On the other hand, LUA achieves even better results than BERT-Biaffine Model. For instance, the F1 score improvements of LUA over it are $10.11\%$ on range $8 - 11$ and $41.23\%$ on range $12 - 24$.

### 4.7 RUNNING TIME ANALYSIS

Table 6 shows the running time comparison among different models. The middle two columns are the time complexity of decoding a label sequence. The last column is the time cost of one epoch in training. We set the batch size as 16 and run all the models on 1 GPU. The results indicate that

---

[3] https://github.com/MiuLab/SlotGated-SLU.

[4] https://github.com/monologg/JointBERT.

[5] https://github.com/juntaoy/biaffine-ner.

[6] https://github.com/cslydia/Hire-NER.

| Model | Theoretical Complexity | Practical Complexity | Running Time |
|---|---|---|---|
| BERT | $\mathcal{O}(n)$ | $\mathcal{O}(1)$ | 5m11s |
| BERT + CRF | $\mathcal{O}(n\|\mathcal{L}\|^2)$ | $\mathcal{O}(n)$ | 7m33s |
| LUA | $\mathcal{O}(n^2)$ | $\mathcal{O}(n)$ | 6m25s |
| LUA w/ Label Correlations | $\mathcal{O}(n^2\|\mathcal{L}\|^2)$ | $\mathcal{O}(n)$ | 7m09s |

Table 6: Running time comparison on the syntactic chunking dataset.

the success of our models in performances does not lead to serious side-effects on efficiency. For example, with the same practical time complexity, BERT + CRF is slower than the proposed LUA by $15.01\%$ and LUA w/ Label Correlations by $5.30\%$.

## 5 RELATED WORK

Sequence segmentation aims to partition a fine-grained unit sequence into multiple labeled coarse-grained units. Traditionally, there are two types of methods. The most common is to cast it into a sequence labeling task (Mesnil et al., 2014; Ma & Hovy, 2016; Chen et al., 2019a) by using IOB tagging scheme. This method is simple and effective, providing a number of state-of-the-art results. Akbik et al. (2018) present Flair Embeddings that pretrain character embedding in a large corpus and directly use it, instead of word representation, to encode a sentence. Liu et al. (2019b) introduce GCDT that deepens the state transition path at each position in a sentence, and further assigns each word with global representation. Luo et al. (2020) use hierarchical contextualized representations to incorporate both sentence-level and document-level information. Nevertheless, these models are vulnerable to producing invalid labels and perform poorly in identifying long-length segments. This problem is very severe in low-resource setting. Ye & Ling (2018); Liu et al. (2019a) adopt Semi-Markov CRF (Sarawagi & Cohen, 2005) that improves CRF at phrase level. However, the computation of CRF loss is costly in practice and the potential to model the label dependencies among segments is limited. An alternative approach that is less studied uses a transition-based system to incrementally segment and label an input sequence (Zhang et al., 2016; Lample et al., 2016). For instance, Qian et al. (2015) present a transition-based model for joint word segmentation, POS tagging, and text normalization. Wang et al. (2017) employ a transition-based model to disfluency detection task, which helps capture non-local chunk-level features. These models have many advantages like theoretically lower time complexity and labeling the extracted mentions at span level. However, to our best knowledge, no recent transition-based models surpass their sequence labeling based counterparts.

More recently, there is a surge of interests in span-based models. They treat a segment, instead of a fine-grained token, as the basic unit for labeling. For example, Li et al. (2019) regard NER as a MRC task, where entities are recognized as retrieving answer spans. Since these methods are locally normalized at span level rather than sequence level, they potentially suffer from the label bias problem. Additionally, they rely on rules to ensure the extracted span set to be valid. Span-based methods also emerge in other fields of NLP. In dependency parsing, Wang & Chang (2016) propose a LSTM-based sentence segment embedding method named LSTM-Minus. Stern et al. (2017) integrate LSTM-minus feature into constituent parsing models. In coreference resolution, Lee et al. (2018) consider all spans in a document as the potential mentions and learn distributions over all the possible antecedents for each other.

## 6 CONCLUSION

This work proposes a novel LUA for general sequence segmentation tasks. LUA directly scores all the valid segmentation candidates and uses dynamic programming to extract the maximum scoring one. Compared with previous models, LUA naturally guarantees the predicted segmentation to be valid and circumvents the label bias problem. Extensive studies are conducted on 5 tasks across 15 datasets. We have achieved the state-of-the-art performances on 13 of them. Importantly, the F1 score of identifying long-length segments is significantly improved.

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
