# OpenReview forum: "Segmenting Natural Language Sentences via Lexical Unit Analysis"
_ICLR.cc/2021/Conference — Reject_

### Official Review · AnonReviewer3 · 2020-10-26
**Segmenting Natural Language Sentences via Lexical Unit Analysis**

**Rating:** 7
**Confidence:** 5

**Review:**

This paper presents a method called LUA, Lexical Unit Analysis for general segmentation tasks. LUA scores all the valid segmentation of a sequence and uses Dynamic Programming to find the segmentation with the highest score. In addition, LUA can incorporate labeling of the segment as an additional component for span labeling tasks.

Pros:
1. LUA overcomes the shortages of sequence labeling as a token-based tagging method and span-based models as well, by treating them separately.
2. The decomposition of scoring label and scoring span allows the pre-computation of the maximum label score for each span, reducing the complexity.
3. This method achieve the state of the art performance on 13 out of 15 data sets empirically.

Cons:
1. The novelty is incremental, as the idea of calculating span-based score and label-based score with DP has been used widely in constituent parsing, which applies interval DP in a similar way. Also check semi-CRF model (Sunita Sarawagi and William W. Cohen, 2004).
2. The way of using neural model to calculate the span-based scoring seems to be very arbitrary (Eq3), without any explanation why it is designed in this way.
3. Label correlations are used to mimic correlation scoring, however the transition between spans are not explicitly modeled.

Questions:
1. LUA is only used in inference stage. Do you think by using LUA in training as well, though slower, the performance can be further improved?
2. Do you have any intuition of why designing the scoring function (Eq3) in that way?

---

> ### Author Response · Authors · 2020-11-16
> **Response to AnonReviewer3**
>
> Thanks for your valuable feedbacks.
>
> Question-1: LAU is only used in inference stage. Do you think by using LUA in training as well, though slower, the performance can be further improved?
>
> Answer-1: DP is used for both training and inference in a consistent manner. As shown in Eq. (8), at training time, we use DP to calculate f(\hat{y}) for inducing the max-margin loss. Besides, in our preliminary experiments, we find that not using DP in training severely degrades the performances. Although DP is involved in training, its running time is still competitive. As shown in Table 6, it is even faster than BERT+CRF by about 1 minute for one training epoch.
>
> Question-2: Do you have any intuition of why designing the scoring function (Eq3) in that way?
>
> Answer-2: The design of Eq. (3) is inspired by the Eq. (14) in ESIM [Chen et al., Enhanced LSTM for Natural Language Inference, ACL 2017]. We will clarify this in the revised version.
>
> Comment-3: The novelty is incremental, as the idea of calculating span-based score and label-based score with DP has been used widely in constituent parsing, which applies interval DP in a similar way. Also check semi-CRF model (Sunita Sarawagi and William W. Cohen, 2004).
>
> Answer-3: Recent methods to text segmentation are dominated by sequence labeling and span-based models. However, they suffer from many problems that are described in Introduction (Section 1). Our proposed LUA solves all of them and have obtained new state-of-the-art results on 13 datasets. Span-based constituent parsers can’t capture the label correlations among constituents and semi-CRF is still an extension of CRF. LUA is highly light-weight and very effective. We strongly believe that LUA will have a great impact on future research.
>
> Comment-4: Label correlations are used to mimic correlation scoring, however the transition between spans are not explicitly modeled.
>
> Answer-4: This is an interesting direction and we will explore it in future work.

---

### Official Review · AnonReviewer1 · 2020-10-28
**Not a clear contribution**

**Rating:** 5
**Confidence:** 3

**Review:**

The paper is well-written, easy to follow and clear. However, the novelty and main contribution of the paper is not clear. The authors used a scoring model to score the composition of each segment, as well as the probability of having a specific label for the segment. The BERT language model is used in the paper to encode the input sequence. The training part is a more like a supervised training and a dynamic programming (DP) approach is used for inference. It is not clear how DP contributes to the success of the model, as the scores for segments are derived during the training (which seems most of the success is coming from the labeled data (i.e. supervised training) and BERT encoding). One other thing about formatting and citing references, some of the references are published in conference proceedings, not sure why authors cited their arxiv version.

---

> ### Author Response · Authors · 2020-11-16
> **Response to AnonReviewer1**
>
> Thanks for your valuable comments.
>
> Comment-1: The training part is a more like a supervised training and a dynamic programming (DP) approach is used for inference.
>
> Answer-1: Actually, DP is adopted for both training and inference in a consistent manner. For example, at training time, we use DP to compute f(\hat{y}) for inducing the max-margin loss (see Eq. (8)). This consistency guarantees that LUA is capable of overcoming the label bias problem.
>
> Comment-2: It is not clear how DP contributes to the success of the model, as the scores for segments are derived during the training (which seems most of the success is coming from the labeled data (i.e. supervised training) and BERT encoding).
>
> Answer-2: There are three important facts: 1) the tasks in text segmentation are all supervised learning; 2) most of our baselines adopt BERT as the sequence encoder; 3) DP is consistently used in both training and inference. Therefore, the achieved promising performances are not attributed to labeled data or BERT but to its attractive properties. As made clear in Introduction (Section 1), our proposed LUA avoids predicting invalid segmentation, overcomes the label bias issue, consistently uses DP in both training and evaluation, etc.
>
> Comment-3: One other thing about formatting and citing references, some of the references are published in conference proceedings, not sure why authors cited their arxiv version.
>
> Answer-3: We will cite the conference/journal versions of these published papers.

---

### Official Review · AnonReviewer4 · 2020-10-29
**This paper presents a new sequential segmentation method by making all possible spans in a sentence as candidate segments and applies dynamic programming to find the best segmentation.  The method is tested with various sequential tagging problems and achieved state-of-the-art results for most of the problems.**

**Rating:** 6
**Confidence:** 5

**Review:**

The paper proposes a new algorithm for sentence segmentation which can be applied to various sequential tagging problems.
The motivation and the description of the algorithm are clearly given, and the proposed method achieved state-of-the-art results for most of the problems and datasets.

The proposed method tries to find all possible segments in the given input sequence, to estimate the scores of the segments using pre-trained BERT representations, and to find the best sequence of segments using the dynamic programming algorithms.  The proposed method is general enough to apply to various sequential tagging problems and natural language sentence analysis.

While the proposed method looks new to apply to the sequential tagging problems in natural language processing, the dynamic programming approach to sequential analysis is a well-known method in the speech recognition community where a sequence of phonemes are segmented into a word sequence.  Also, a similar method has been applied to the segmentation of character sequences into word sequences for the languages that have no delimiters between words, such as Chinese and Japanese.   In these views, the novelty of the paper is not high.   On the contrary, the application of the BERT-based representation to the sequence segmentation tasks such as sentence segmentation and sequential labelling may be new, and the finding that this method can attain a state-of-the-art performance in those problems could be worth reporting.

---

> ### Author Response · Authors · 2020-11-16
> **Response to AnonReviewer4**
>
> Thanks for your valuable comments.
>
> Comment-1: While the proposed method looks new to apply to the sequential tagging problems in natural language processing, the dynamic programming approach to sequential analysis is a well-known method in the speech recognition community where a sequence of phonemes are segmented into a word sequence. Also, a similar method has been applied to the segmentation of character sequences into word sequences for the languages that have no delimiters between words, such as Chinese and Japanese.
>
> Answer-1: Would you please show us the related works in speech recognition? We will clarify the differences between them and our work. The proposed LUA has made significant contributions to NLP. Currently, the dominant approaches to text segmentation are sequence labeling and span-based models. However, they suffer from many problems, which have been made clear in Introduction (Section 1). LUA addresses all of them and, importantly, it has established new state-of-the-art performances in 13 datasets. LUA is very light-weight and highly effective. We strongly believe that LUA will be a new framework for text segmentation. We sincerely hope the reviewer can see the value of this work to NLP.
>
>
>
> Comment-2: On the contrary, the application of the BERT-based representation to the sequence segmentation tasks such as sentence segmentation and sequential labelling may be new, and the finding that this method can attain a state-of-the-art performance in those problems could be worth reporting
>
> Answer-2: In fact, most of the compared baselines adopt BERT as the sentence encoder, but our proposed LUA has still significantly outperformed them. This demonstrates that the superiority of LUA is not attributed to BERT but thanks to its attractive properties. As described in Introduction (Section 1), LUA avoids predicting invalid segmentation, overcomes the label bias issue, consistently uses DP in both training and evaluation, etc.

---

### Decision · Program_Chairs · 2021-01-07
**Final Decision**

**Decision:**

Reject

**Comment:**

This paper is concerned with sequence segmentation. The authors introduce a framework which they call 'lexical unit analysis' - a neural network is used to score spans and then dynamic programming is used to find the best scoring overall segmentation. The authors present extensive experiments on various Chinese NLP tasks, obtaining better results than the systems they compare to.

Reviewers raised concerns, including about novelty. In my view, beyond beating the state of the baselines on the chosen tasks, it is hard to extract an actionable insight or novel conceptual understanding. Therefore, the paper is not recommended for acceptance in its current form.